

# Popigai and Chicxulub craters:
# multiple impacts and their associated grabens

Jaroslav Klokočník[1], Václav Cílek[2], Jan Kostelecký[3,4], Aleš Bezděk[1]

[1]Astronomical Institute, Czech Academy of Sciences, CZ 251 65 Ondřejov,
Fričova 298, Czech Republic, jklokocn@asu.cas.cz; bezdek@asu.cas.cz
[2]Geological Institute, Czech Academy of Sciences, Prague, CZ 165 00 Praha 6, Rozvojová 269, Czech
Republic, cilek@gli.cas.cz
[3]Research Institute of Geodesy, Topography and Cartography, CZ 250 66 Zdiby 98, Czech Republic
[4]Faculty of Mining and Geology, VSB-TU Ostrava, CZ 708 33 Ostrava, Czech Republic,
kost@fsv.cvut.cz

*Correspondence to*: Jaroslav Klokočník (jklokocn@asu.cas.cz)

**Abstract.** More advanced data (gravity field model EIGEN 6C4 with GOCE gradiometry data instead of EGM2008) and a more sophisticated method (using a set of the gravity aspects instead of the gravity anomalies and the radial second derivative of the disturbing potential only) enable a deeper study of various geological features, here the impact craters Chicxulub and Popigai. We confirm our results from 2010, extend them, and offer more complicated models, namely by means of the gravity strike angles. Both craters are double or multiple craters. The probable impactor direction is from NE for Chicxulub and SE-NW for Popigai. The both crater formations seem to be associated with impact induced tectonics that triggered development of impact grabens.

## 1. Motivation

In 2010, we published (in this journal) results of our tentative analysis of the gravity data for two

areas of proven, huge impact craters Chicxulub and Popigai, based on the global combined gravity

model EGM 2008 (Pavlis et al. 2008 a,b, 2012) till degree and order (d/o) = 2159, top of the

modelling of the gravity field of the Earth at that time as for precision and resolution. We suggested

(Klokočník et al., 2010) that Chicxulub (north Yucatán, México) may be a double crater and

Popigai (north Siberia, Russia) may be a multiple crater [see the labels "Popigai I-IV", according

to Rajmond's catalogue (2009), on © Google Earth]. The craters II-IV are not yet proven impact

crater candidates. Altogether they would create a hypothetical Popigai crater's family, a catena

(see Supplement S3).

Since that time we have analysed many geological features on the Earth, the Moon and Mars,

we make use of new gravity models and other data sources (see below) and we summarized our



results in three books and about 15 papers  (e.g., Klokočník & Kostelecký 2015, Klokočník et al.,
2017, 2018, 2020, 2021, 2022 a,b, 2023 a,b).

3        We (and the readers) are well aware that solely the gravity data are not unambiguous to detect

ground density anomalies (causative bodies); we always need and seek for additional data
(geological, geophysical, seismic, topography, archaeology). Therefore, with our data and method,
we offer only a step to possible field explorations and subsequent interpretations but not a
confirmation of the structures.

8        With increasing precision, accuracy, resolution and reliability of our knowledge about the

gravity and magnetic fields, we can test diverse applications impossible before. This is not only
about data, it is also about methodology; the traditional gravity anomalies are no longer sufficient.
We apply a set of the *gravity aspects* instead (Sect. 2 and Supplement S1).

12       Another impetus for this study was similarity of the two craters in the sense that they are directly

associated with close linear structures that seemingly have nothing to do with the impact event,
but occur closely to the craters. This could be a coincidence of two different, genetically
independent geological phenomena, indeed, or a sign of existence of a trench being modified by
the impact event. The impact shook affects the entire region and those previously existing faults
or fault zones that were in an extensional tectonic regime were activated to form impact graben.

18       This paper is a revival concerning our previous findings about *Popigai* and *Chicxulub*. Now we

have (in comparison with Klokočník et al., 2010) better tools (the set of the gravity aspects) and a
better gravity model (EIGEN6C4, with GOCE data), thus we can support or reject those older
results. We offer new and hopefully more convincing results in favour of a double/multiple
character of both Chicxulub and Popigai craters. These results are not in conflict nor with known
geology of the areas nor with the information from magnetic intensities (not studied here).

24       Many figures are gathered in *Supplementary materials* in S*i*: there is S2 tutorial, with tests about

artefacts, S3 for Popigai, and S4 for Chicxulub. The theory is shortly repeated in S1. See:
www.asu.cas.cz/~jklokocn/Popigai_Chicxulub_2024_supplements/.
**2.   Notes on theoretical preliminaries**
Gravity (gravitational) aspect (descriptor) is a functional/function of the disturbing gravity
gravitational field potential $T_{ij}$. We work with the gravity anomaly (or disturbance) $\Delta g$, the Marussi



tensor ($\Gamma$) of the second derivatives of the disturbing potential ($T_{ij}$), two gravity invariants ($I_j$), their
specific ratio ($I$), the strike angles ($\theta$) and the virtual deformations ($vd$).
The theory came mainly from Pedersen & Rasmussen (1990) and Beiki & Pedersen (2010).
The theory with examples is summarized in our books (Klokočník et al., 2017, 2020, 2022b); it
cannot be (due to space reasons) repeated here (see Supplement S1). Only a few notes to the theory
follow.
The gravity aspects are sensitive in various ways to the underground density contrasts
(variations) due to the causative bodies, exciting the relevant gravity signals. The set of the gravity
aspects tells us much more about the causative body than the traditional $\Delta g$ only; it informs about
the location, shape, orientation, a tendency of the ground structure to 2D or 3D patterns, stress
trends and may partly simulate a "dynamic information" about tensions although the input data are
always the same – the harmonic geopotential coefficients of a *static* gravity field model
For example, the strike angle $\theta$ mathematically can be the main direction of the Marussi tensor
$\Gamma$ of the second derivatives of the disturbing potential (the first column and first row of $\Gamma$ is
identically equal zero for this preferred direction). The strike angle is, from geophysical point of
view, a direction important for description of the ground structures. It may indicate areas with a
lower density or higher porosity or a "stress direction" or both or the areas under a strong influence
of rapid and/or intensive geomorphic processes. When $I=0$, the values of $\theta$ may be symptomatic
of flat causative body. For more details see Beiki & Pedersen (2010) and our S1.
A usual situation is that the strike angle $\theta$ has diverse directions, as projected on the Earth'
surface. The *combed strike angles* are the strike angles oriented roughly in one and the same
direction in the given area. Theory for the "combed" strike was explained, together with relevant
statistics, in Klokočník et al. (2019). For statistical use we defined a degree of alignment of the
strike angles by the "comb coefficient" *Comb* as a relative value in the interval ⟨0,1⟩, where 0
means that the strike angles are „not combed" (totally dishevelled, the vectors $\theta$ are in diverse
directions) and *1* means "combed" (perfectly kempt, the vectors of $\theta$ are oriented into one
prevailing direction). If *Comb* is smaller than 0.55, we say that $\theta_i$ of the given region are "not
combed"; if *Comb*>0.65, we say that $\theta_i$ are "combed", and for *Comb*>0.99, they are perfectly
aligned. The alignment may take a linear form (e.g. along a fault) or can shape a halo (around
craters); see Theory in S1 and many examples in the tutorial S2.



## 3. Data, computation, and figures

We always start all our computations with the harmonic geopotential coefficients (Stokes parameters) of the static global gravity field models as the input data; they describe the gravitational potential of the Earth. The whole theory is prepared in such a way that we cannot use another input than the harmonic coefficients.

We make use of a high resolution combined *E*uropean *I*mproved *G*ravity model of the *E*arth by *N*ew techniques (*EIGEN 6C4*, Förste et al., 2014), expanded to degree and order (d/o) 2190 in spherical harmonics; this corresponds to the ground resolution 5x5 arcmin or ~9 km on surface. Precision of EIGEN 6C4, expressed in terms of *Δg*, is *N*=10 mGal, but in many civilized land areas and over the oceans and open seas is much better. The authors of EIGEN 6C4 have not access to most of the recent high resolution terrestrial gravity data on the continents, thus they took a synthesized gravity anomaly grid based on EGM2008 (Pavlis et al., 2008 a,b, 2012). That means that the errors for high d/o terms in EIGEN 6C4 are dominated by the relevant errors in EGM2008. To estimate the precision for the given area of interest, not only a general figure 10 mGal, one needs to inspect gravity anomaly commission error maps of EGM2008 (Pavlis et al. 2008 a,b; also in S3). For the northern Yucatan peninsula, we get *N*=4-8 mGal, for Popigai in Siberia a bit worse.

Note about other data sources: *ETOPO 1* global surface topography (Amante and Eakins, 2009), a global 1' relief model of the Earth surface that integrates land topography and ocean bathymetry from a large number of satellite and other measurements. Its precision globally should be 10 m in heights, its accuracy ~30 m. There are alternative topography data files. Google Earth is also helpful. *Bedmap 2* is a subglacial topography valid for Antarctica (Fretwell et al., 2013). It contains the bedrock elevation beneath the grounded ice sheet. It is given as a 1x1 km grid of heights of the bedrock above sea level, but actual measurements are often much sparser.  We also worked with the *RET 14* (Hirt et al., 2016), a degree-2190 gravity field model *SatGravRET2014*, given as a set of harmonic geopotential coefficients*,* meaningful only for the continent of Antarctica (not globally!). Roughly speaking, it combines the global gravity field model *EIGEN 6C4* and the Bedmap2 topography.

The data for magnetic analysis on the Earth are the grid value from the worldwide EMAG 2 model for magnetic intensities (Maus et al., 2009). There are also gravity filed models, global topography, and magnetic data for the Moon and Mars; gravity field models of the Moon and Mars provide already sufficient ground resolution for our analysis; it is about 10 km for the Earth and





the Moon, but only 130 km for Mars. Here, we have some examples of the results based on these models in S2.

We computed the gravity aspects over many regions of the world in a step 5x5 arcmin in latitude and longitude, corresponding to the ground resolution 9 km. But we can also use (and use here) a 4 km resolution without any degradation of the results (we offer some results of our truncation error tests and testing of artefacts in Klokočník et al., 2021 and here in Sect. 4 and S2). This higher resolution sometimes adds a new and valuable information.

The numerical stability of computations of high degree and order functions in the aspects is extremely important; it was intensively investigated, tested and is guaranteed to much higher degree and order than we need here (work done during last about ten years by the co-authors of this paper, plus Sebera et al. 2013 and Bucha & Janák, 2013).

Our figures are not generated by an automat, but created manually and individually with specific scales to emphasize various features and details. We plot all the quantities in geodetic (geographic) latitudes and East longitudes.

The gravity disturbances (anomalies) are given in milliGals [$mGal$], the second order derivatives are in Eötvös [$E$]. Let us recall that $1 mGal = 10^{-5} ms^{-2}$, $1E \equiv 1$ Eötvös $= 10^{-9} s^{-2}$. The invariants have units $I_1$ [$s^{-4}$] and $I_2$ [$s^{-6}$]. The strike angle $\theta$ [$^o$, $deg$] is expressed in degrees with respect to the local meridian; its red colour means its direction to the east and blue to the west of the meridian. Often, we plot $\theta$ black and white only. The strikes are shown in a regular grid 5x5 arcmin; it has not any geophysical meaning, this is just the choice for plotting.

## 4. Artefacts

### 4.1. Our previous work

To avoid various misinterpretations we need to test the input data to our analyses in various ways (e.g., Klokočník et al., 2021). We made our best to avoid the artefacts, but nevertheless, they cannot be excluded (S4: slides # >23). The important facts are the resolution and statistical significance.

The *ground resolution* (GR) of the gravity field model is derived from maximum d/o of its spherical harmonic expansion (the definition of GR is in S2: 23), more in Sect 4.2.

Another important factor is the signal-to-noise ratio $R=S/N$. The "signal" $S$ is given as the range of gravity anomalies in the area of interest. The noise $N$ is the commission error of the gravity



anomalies $\Delta g$ (see figure in S1, last page) or the estimated precision of $\Delta g$ of EIGEN 6C4. We
need $R>3$ to have statistically significant results. We have:

3       min $R$ = (min (max| $- S$|, max + $S$))/(max $N$);

4       max $R$ = (max (max| $- S$|, max + $S$))/(min $N$).

With the figure from S1, defining $N$, figures below and in S3 and S4, defining $S$, for $\Delta g$ of Popigai
and Chicxulub, we get $R$(min,max) = 8–15 for Popigai and $R$(min,max) = 5–20 for Chicxulub.
**4.2. Resolution**
The reader certainly knows about the "canals" or "human faces" on Mars; they disappeared with
better new observations and higher resolution (S2: slide #25). The adequate GR of the gravity
model is an important, necessary but not sufficient condition and a limiting factor for correct
interpreting the gravity aspects. The definition of GR is recalled in S2:23 and we can only repeat
(Sect. 3) that the GR of EIGEN 6C4 is 9 km but can be enhanced to about 4 km (see above).  This
provides clear limit for any interpretation. The subglacial topography has a similar problem: data
gathered from airplanes over Antarctica (Bedmap2) are not homogeneous in latitude and longitude,
not complete and with large gaps (Fretwell et al., 2013). Taking the resolution of the subglacial
topography data Bedmap2 as published, i.e. net 1x1 km literally, we can get pictures of the
topography showing unbelievable shapes instead of real features (Klokočník et al. 2021); the
artefacts are looking like walls, pyramids, etc (for example S2: 28). The problem is that the data
density is somewhere ~5 km but ~50 km on other localities; there are zones with no data at all. We
have to know how well by the data is covered the area of our interest (*fig. 3* in Fretwell et al., 2013).
**4.3. Signal degradation and truncation error tests**
A treacherous situation with artefacts can be demonstrated by using the gravity model to its
maximum degree and order d/o in harmonic expansion, exactly as it was published. The result may
be surprising. A model is published say to d/o =1200 but recommended to be used (by the authors
of the model themselves) only to max d/o = 600. The reason is stabilization of the large matrix
inversion by Kaula rule for the higher-degree part of the model. The full model can show a
significant graining in the gravity aspects leading to total damage of the signal, see S2 (in all the
gravity aspects; faster degradation was observed for he gravity aspects with higher derivatives of
the disturbing potential; Klokočník et al., 2021).



Figure S2:29 shows one of our many tests, in this case for the Moons' crater Copernicus with
the gravity model GRGM1200A (Lemoine et al., 2014) till d/o = 1200. Practically useful limit at
d/o ~ 600 corresponds to the theoretical ground resolution ~10 km. This is already comparable to
the Earth, to its EIGEN 6C4 gravity model to d/o=2190 (Foerste et al., 2014) because the Moon is
smaller than the Earth. When we use GRGM1200A up to d/o=600, we can see a reasonable result
S2:29 showing all known features. When we cut at d/o 130, a part of useful signal is lost. When
we use the model to d/o=1200, graining is significant and we can interpret nothing.
Let us imagine that today we know the gravity field of the Moon only to d/o=10. What
information we lose (or is „hidden") in a comparison with the full model to d/o=600? Not only the
resolution of the former is much lower (expected) but sometimes artefacts are created, look at
S2:30 (expected?). Only a further gravity field improvement would eliminate such artefacts. We
are now in analogical situation with the gravity field EIGEN 6C4 to d/o=2190 for the Earth. What
we would lose and which artefacts might be generated with, say, the model cut at d/o=80? The
slides S2: 31, 32 show the result in terms of the strike angles.  Often the basic trend in both full
model and the cut model is the same, but not always; thanks to the dramatic difference in maximum
d/o-used, it must be expected, but, in any case, it is a warning. The artefacts „lurk" and can
eventually attack and distort our endeavour concerning the geointerpretations. It is not probable
but not excluded even for EIGEN6C4 to 2190; the case of artefacts due to an aliasing of the gravity
aspects on Sahara is in *fig. 5a* in Klokočník et al. (2021) and S2: 33 here.
**5.  Popigai**
**5.1. Introductory notes and geology**
This large, proved, exposed impact crater Popigai/Popigaj is in Russia near Khatanga (Chatanga,
port on river), Krasnoyarsk district, Siberia (geodetic latitude and longitude of the centre of the
crater: $\varphi=71°36'$N and $\lambda=110°55'$E). It is a 100-kilometre diameter crater ~35 million years old
(from the late Eocene epoch). It was considered for the first time as an impact crater by Masaitis
et al. (1972); it was based especially on petrographic observations on the various breccias. It is the
largest known impact crater post-dating the Cretaceous–Tertiary boundary (e.g., Vishnevsky &
Montanari, 1999, Whitehead et al., 2000, Koeberl 2009, Masaitis 1998, Masaitis et al., 2019).
The impactor is suggested to have been a H chondrite asteroid several kilometres in diameter
(e.g., Schmitz et al., 2015) from the main asteroid belt. The asteroids may have approached the





Earth at comparatively low speeds, passed the Roche limit and produced a meteoritic shower. But
also a multi-type asteroid shower may have been recorded, triggered by changes of planetary
orbital elements due to orbital resonances (see again, e.g., Schmitz et al., 2015). There is no
agreement among researchers.
The Popigai crater lies on the eastern edge of the Archean Anabar Shield, which is mainly
composed of granitoids and gneisses. It is surrounded by a relatively complex envelope of
Precambrian, Paleolozoic and Cenozoic rocks, which reach 1-1.5 km in thickness at the point of
impact (Masaitis 1998, Pilkington et al., 2002). It is a multi-ring structure with three concentric
rims visible.
The bedrock is crushed to depths of at least 5 km according to the results of drilling and
geophysical measurements. The internal structure of the crater is quite unusual and contains a
number of enigmatic phenomena, such as the presence of impact breccias fused into glassy, also
fragmented tagamites (breccia within breccia). Vishnevsky & Montanari (1999) propose that the
contrasting sedimentology or the presence of water in some layers of the original pre-impact
sedimentary succession may have triggered a whole chain of impact phenomena. A similar result
could be caused by the nearly simultaneous, close impact of two or more meteorite fragments.
The long-term evolving terrains always have a complex tectonic framework, or rather a
sequence of tectonic regimes creating a network of faults of different ages and directions. Masaitis
(1998) in his diagram of the crater shows radial tectonics in the immediate vicinity of the crater,
while in Vishnevsky & Montanari (1999), long faults of NW-SE and SW-SE directions are
displayed. Somewhat unexpectedly, faults in the NS direction predominate in the crater itself,
without any apparent influence of the impacting body.
Looking at the broader tectonic framework, we see a number of significant structures based on
faults of approximately NS direction. The latter follows the Ural Mountains, the western margin
of the Central Siberian Plateau, the Verkhoyansk Chrebet, and some rivers such as the Daldyn
River directly in the crater and around parts of the course of the Anabar and Malaya Kuonamka
rivers. Perpendicular to them, a long EW structure visible with ETOPO 1 (Fig. 1a), is located,
north of the crater (see the arrows from E and from W).
The impact's shock pressure instantaneously transformed graphite in the ground into
diamonds (e.g., Masaitis et al., 1972, Masaitis 1998, Deutsch et al., 2000). The aggregates of
diamonds are sometimes up 1 cm large. They tend to retain the appearance of graphite or original



organic aggregates. They are bound to outcrops of original rocks with an admixture of graphite or
coal substance. They are absent in the central part of the crater, where the pressure and temperature
were too high for diamonds to form or preserve (see, e.g., *fig. 1* in Masaitis 1998). Vishnevsky and
Montanari (1999) presented a diamond occurrence map (their *fig. 6*, p. 26) showing a more or less
chaotic distribution caused by both an irregular admixture of carbon-rich impacted rocks and a
complex, multiphase crater evolution. Popigai is most probably linked to ejecta horizons occurring
in marine sequences of Late Eocene age.
Pilkington et al. (2002) presented $\Delta g$ based on the local GETECH Ltd gravimetric data
showing a negative "valley" going from the main and proven Popigai crater in the SE direction,
their *fig. 3a*, which is an indication of a possibility that we deal with double/multiple craters.
Popigai may be a multiple crater, a catena (Figs. 1b,c and slides #6-21 in S3); it was proposed
in Klokočník et al. (2010), based on analysis of $\Delta g$ and $T_{zz}$ derived from (at that time the best)
global gravity field model of the Earth EGM2008. Popigai may represent one of two or three
simultaneous impacts from one original asteroid. Some authors consider asteroid shower from a
single parent-body breakup (Schmitz et al., 2015). For notes about binary asteroids see Sect. 6.1.
Popigai has been designated by UNESCO as part of the world's geological heritage. Due to
economic reasons, exploration work in this quite remotely area (the joint German-Canadian-
Russian expedition) has ceased before 2000 (according to Deutsch et al., 2009).
For completeness of these records, we note that near Popigai, roughly in the SW direction of
Popigai, another, not yet proven crater, independent of and bigger than Popigai, the feature known
as Kotuykanskaya, a hypothetical impact crater, is located (see Rajmon 2009; $\varphi=69°30'$N;
$\lambda=100°25'$E; Klokočník et al. 2020c and references in this paper; see also S3: 14-17 here).
Now, we have (in comparison with 2010) better tools (the set of the gravity aspects) and better
gravity model (EIGEN6C4, with GOCE data), thus we can support or reject our older results. We
will argue in favour of a multiple crater.
**5.2. Our new gravity results for Popigai**
The recent global satellite-based surface topography depicted by the ETOPO 1 model is shown in
Fig. 1a (and variants in S3). There is a broad topographic low in NE, E, and SE directions from
the main Popigai crater. One reckons that the terrain may has been strongly affected by water/ice
erosion and other influences since time of the impact event – e.g., a river is flowing throughout the



1    bottom of the crater, the rim is disrupted significantly on two places with consequences on the

2    gravity signal (see reaction in the gravity signal in the following figures).

a

**Figure 1a**: **Popigai:
ETOPO 1 topography [m],
shaded relief; red dot
means the crater's centre;
an alternative projection,
with contour lines, is in S3.
The arrows show EW
linear structure; its gravity
signal is weak.**

20    b                                        c



1    **Figure 1b: Δ*g* [mGal] and *θ* [deg], *I* < 0.9 with the ETOPO 1 topography**

2    **Figure 1c: $T_{zz}$ [E], strike angles *θ* [deg], and the topography from ETOPO 1 [m].**

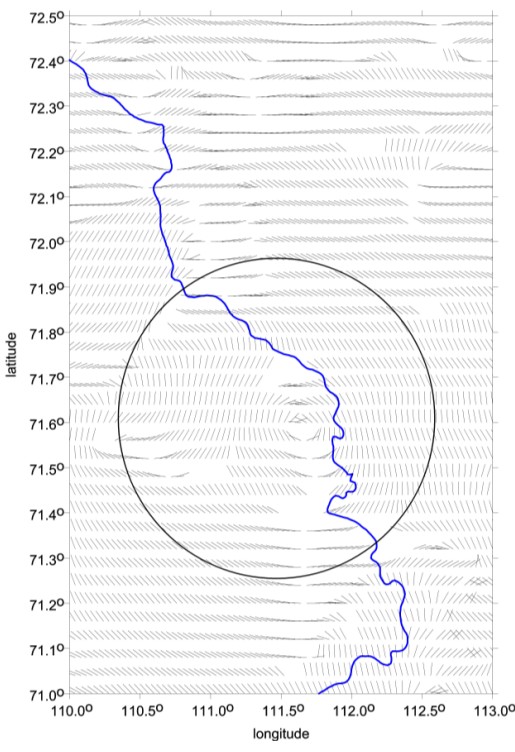

d

**Figure 1d: details for *θ* in the main, largest and proven Popigai crater. The halo of the strike angles combed around the crater bottom (circle) and its central. The Popigai river (in blue) locally disrupts the halo.**

The structure is characterized by a strong gravity anomaly low of Δ*g*=-40 mGal and $T_{zz}$=-30 E
amplitude (EIGEN 6C4). Superimposed on the gravity low is a concentric ring-shaped high,
fragmented now due probably to postimpact evolution. The central peak is visible, Figs. 1b-d, but
it is not too intensive.
Fig. 1b shows Δ*g*, Fig. 1c presents $T_{zz}$. Fig. 1d is a zoom just for *θ* (*I*<0.9) in the main crater,
with a halo of the strike angles; there is a mark of the central peak, too. The topography (Fig. 1a)
and the gravity aspects (Figs. 1b-d and S3) do not correlate. Beside the main, proven crater, we
clearly see more candidates for impact craters (which are a bit smaller than the main crater). They
are lined in the SE direction (Klokočník et al., 2010, 2020b, Khazanovitch-Wulff et al., 2013). It
is obvious from Figs. 1b-c, from broad negative Δ*g*, from negative belts and semicircles of $T_{zz}$,
and from the strike angles *θ* included in these figures (also S3: 8, 9, 21). These *θ* have tendency to be
directed along the long axis of the whole Popigai family (SE-NW), interrupted only locally inside



the potential craters (e.g. S3: 21). We labelled these crater candidates as Popigai II-IV in
(Klokočník et al., 2010). Counting from the main and proven Popigai crater (Popigai I), a large
circular structure is visible on the SE rim of the main crater. It can be the companion crater – what
we called Popigai II in (Klokočník et al., 2010). At that time, we had not the strike angles and the
virtual deformations at our disposal; with them now, we can demonstrate better that Popigai can
indeed be a double or multiple craters, i.e. catena, a rare phenomenon on the Earth (the "Popigai
family").
**6.  Chicxulub**
**6.1. Introductory notes and geology**
The impact crater Chicxulub (Northern Yucátan, México) is centred beneath Chicxulub village
($\varphi$=21°17′N and $\lambda$=89°30′W) near the Progreso port. The crater is huge, not exposed, with a
diameter 170-250 km, and about 65 mil years old. This enormous impact represents an external
forcing event with far-reaching, global impact in mass extinction, as is well-known (the KT event).

16       The Yucátan peninsula is a low-lying limestone platform. The crater is buried under Quaternary

carbonate sediments (0.6-1.0 km thick), lying over Tertiary sandstone and volcanic rocks. The
northern (nearly)-half of the now-buried crater is in shallow waters of the Sea of Campeche (of
Gulf of Mexico), which then falls, at the northern end of the Campeche Bank, to deeps in the
Campeche Escarpment (fault).

21       The origin of the impactor in the Solar System is not yet clear. Bottke et al. (2007) proposed

that the Chicxulub impactor could have originated from a moderately young asteroid family
Baptistina. Located in the inner main belt of asteroids, this cluster is favourably positioned to
deliver large objects (>5 km) to the terrestrial planets. A recent analysis of Nesvorný et al (2021)
claims that the crater was produced by impact of a carbonaceous chondrite and suggest that the
impactor came from a main belt asteroid that quite likely ($\simeq$ 60% probability) originated beyond
2.5 au. Some authors consider a comet as the impactor. The impactor might be also a binary
asteroid, but it is rare to achieve two craters with two asteroids. The asteroids must be sufficiently
separated (s/c "wide binaries"). Two closer impactors can produce one crater, one elongated crater,
or two overlapping craters (Miljikovic et al., 2013).



The impactor direction has been studied among others by Hildebrand et al. (2003). We quote:
„the impact direction was towards the northeast based on the asymmetries preserved in various of
Chicxulub's structural elements in addition to the vergence observed in the central uplift:
compressional structures outside the crater rim, the rim uplift, compressional deformation
preserved in the slumped blocks, morphology of the peak ring, off centre position of the central
uplift in the collapsed disruption cavity (CDC), elongated CDC, and initiation of slumping of
Cretaceous stratigraphy off the Yucatan platform."
The literature about the Chicxulub crater is really rich: from Alvarez et al. (1979, 1980); Smit
and Hertogen (1980); Hildebrand et al. (1990, 1995); Ramos (1975) …. to Campos-Enriquez et al.
(2004), Gulick et al. (2008, 2016); Goderis et al. (2021) or Urrutia-Fucugauchi et al. (2022), and
many more. This is not a review paper to mention all. We recall the important role of terrestrial
gravity data in its study. Already Hildebrand et al (1998) used not only the terrestrial gravity
anomalies (measured for oil/gas prospection) but also horizontal second derivatives to enhance
resolution; but they did not know the concept of the gravity aspects.
According to Klokočník et al. (2010), Chicxulub may be a double crater; it was suggested after
the analysis of $\Delta g$ and $T_{zz}$ based on EGM 2008 (compare with *fig. 2* in Hildebrand et al., 2003). In
this paper, we present our new results: we further support this hypothesis, with EIGEN 6C4 gravity
model (better data) with all the gravity aspects (more sophisticated tools).
Strong impacts like this one have global effects; regionally the enormous pressure can trigger
many postimpact activities and features. Let us recall Donofrio (1998) who wrote: "Seventeen
confirmed impact structures occur in petroliferous area of North America, nine of which are being
exploited for commercial hydrocarbons… Disrupted rocks in proximity to impact structures, such
as Chicxulub in the Gulf of Mexico off Yucatan, also contain hydrocarbon deposits". James et al
(2002), p. 40, wrote: "…There are several craters that host fossil fuels, with the submarine
Chicxulub impact crater…" and "…a total of 21 craters have oil/gas/hydrocarbon/coal resources,
of which 19 host oil and gas." The reader can see slides #17-18 in S4.
A rapid burial of Chicxulub by Cenozoic sediments contributes to its preservation but also limits
its study. The direct, surficial or submarine geological study of Chicxulub is impossible because
the structure is buried by several hundred metres to 1 km of porous Tertiary limestones (Ramos
1975). A 2016 drilling project revealed a central ring composed of originally deep-seated, coarse-
grained granite (Morgan et al., 2016). It is important because analogously we can expect rocks





from depths of >10 km in, for example, lunar craters, as Kring (2016) reports for the crater
Schrödinger. The concentric structure of Chicxulub is surrounded by the ring of cenotes. It
indicates finely fractured, and more permeable zones, on those the extensive cave systems
developed. At the surface it manifests as cenotes i.e. collapsed cave ceilings (Perry et al. 1995). In
a wider surrounding, the karst phenomena are known on the north-eastern margin of the Yucatan
in the Holbox tectonic zone, but here they are much more likely connected to the broad active arc
that encircles Cuba from the north and trends toward the Yucatan (the Pinar Zone and Oriente
Fault Zone).
**6.2. Our new gravity results for Chicxulub**
The gravity anomalies around Chicxulub are shown in Figs. 2b-d, the radial component $T_{zz}$ in
S4:21, $T_{xx}$, $T_{yy}$, $T_{zz}$ in S4:22, the invariants $I_1$, $I_2$ in S4:23, their ratio $I$ in S4: 24, and $vd$ in Fig. 2e
and S4: 24, 26, 27. The strike angles $\theta$ are in S4:25; they are also underlying several other figures
with the gravity aspects. We do not forget on the ring of cenotes (sinkholes, originally potable
water sources used by Maya; S4: 9-11, 27, and 31).
The surface topography ETOPO 1 (Fig 2a) does not correlate with the gravity aspects (the crater
is not visible on the surface); even Ticul Fault and Ticul Sierra (hills) do not correlate with gravity.
We can see the positive $T_{zz}$ at the central peak and along the rims, negative $T_{zz}$ in between the
rings. The strike angles are combed inside the crater, clearly laid down along the rims (analogy to
Vredefort, S2: 9), so they correlate also with the ring of cenotes (S4: 10). Outside the central
crater, the prevailing direction is SW-SE. The strike angles, combed around Chicxulub to halos,
following the craters' rims, are strong on land. The ring of cenotes agrees well with the halo created
by the strike angles along the outer, most compact ring of the crater. Cenotes then continue like a
cluster on the east edge of the crater (S2: 9).
Tertiary sedimentary layers of the flat northern Yucatan outside the crater have, as expected,
linear and also highly combed $\theta$. A contrast of the density of sediment or a changed porosity (with
respect to surrounding rocks) is high enough to be gravitationally distinguishable. The cenotes as
well as oil/gas deposits near Yucatan, although epigenetic, are not there by a chance (e.g., Grieve
2005, p. 21) but as a consequence, direct or indirect, of the impact event.



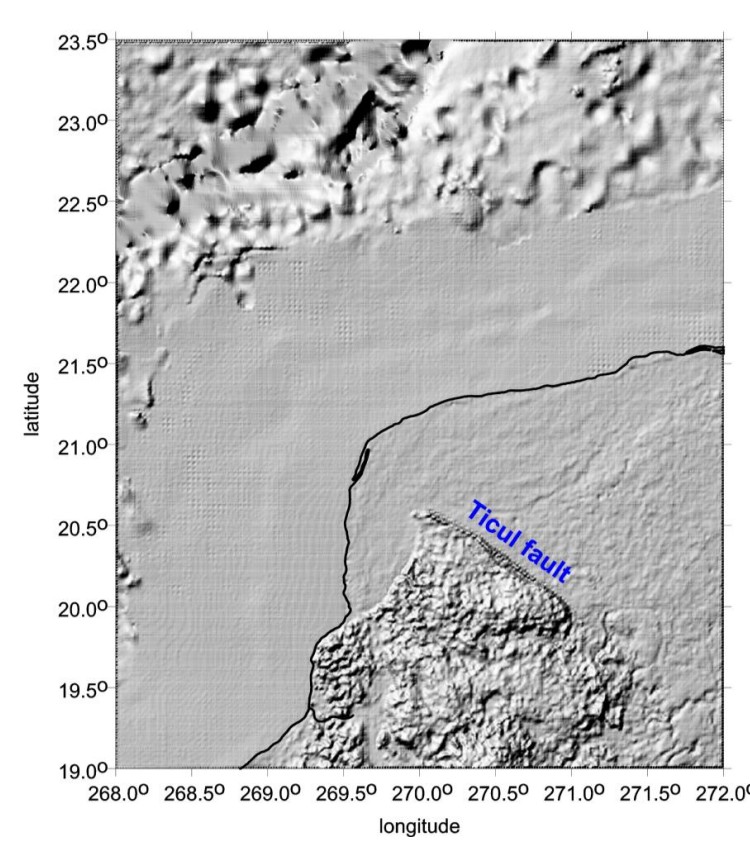

**Figure 2a: Northern Yucátan, México, flat low-land and shallow-sea area of the buried Chicxulub impact crater, ETOPO 1 topography (without any gravity aspect added) in an enhanced shaded relief scale (compare to S4: 8-10, 26). Tertiary sediments cover the impact crater; only semi-circular "shadows" due to the cenote ring are observable (here and in S4: 8). Black line: the coast.**

Our figures demonstrate a halo around the central part (min. two rings). The strike angles are also strongly linearly combed far from the crater, mainly SW-NE (due to the local high porosity around and the cenotes outside the rims of Chicxulub E of them).

Fig. 2e presents the virtual deformations (*vd*), red for dilatation, blue for compression. The *vd* perfectly depict the bottom of the crater, its central peak, the rings, and the combed areas around.

We newly analysed the negative "southern gravity anomaly" (located S to SW of the main crater) in the NS direction; we call this feature the "tail", see Figs. 2b-d, S4: 15-17, 25, and 26. The prevailing, standard opinion is that this is a pre-impact feature (e.g., Gulick et al., 2008; Urrutia-Fucugauchi et al., 2022).



The tail or trench-like structure or NS elongated depression of the graben type has a negative
gravity anomaly. Linearly combed strike angles in the same direction (Fig. 2b,c and S4) indicate
syngenetic feature with the impact crater(s). The *vd* in Fig. 2e show the best the whole linear
feature, the hypothetical impact graben, connected with the impact craters. The southern tail would
be its southern end.

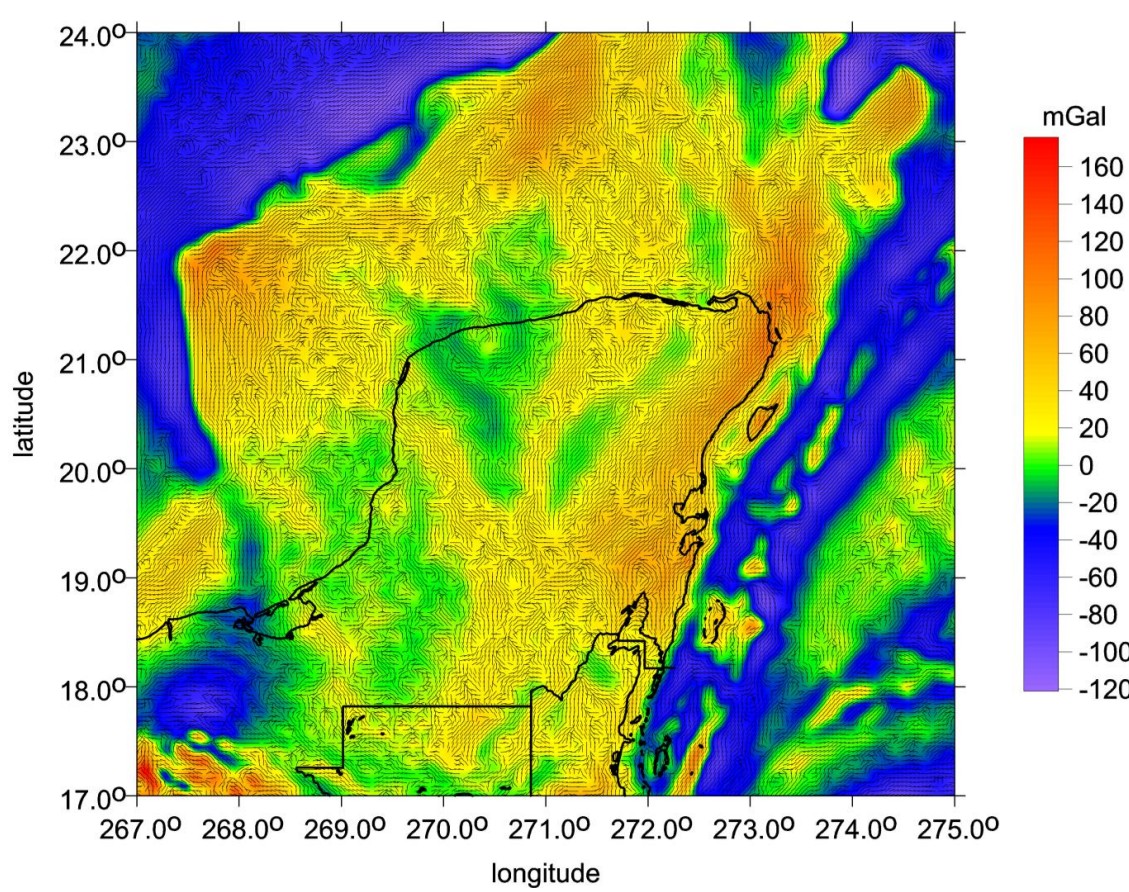

**Figure 2b: Northern Yucátan, México, the area of Chicxulub impact crater, using the gravity**
**model EIGEN 6C4 to maximum degree and order d/o=2190, with a 4 km resolution. The gravity**
**anomalies $\Delta g$ [mGal], together with the gravity strike angles $\theta$ [deg], $I < 0.9$. Black lines: coast and**
**state borders. The strike angles as a parameter of the gravity anisotropy tensor $\Gamma$ reveal up to**
**three ringed structures of the Chicxulub basin. The combed strike angles correlate with oil/gas**
**deposits (it continues to SW to Campeche off-shore oil fields), also with its rims and with a**
**(semi)ring of the cenotes (on land). These are sinkholes (karst features) in the local limestone**
**sediments; they were used by Maya as a source of drinkable water. They represent one of the**
**post-impact effects. The second radial derivative $T_{zz}$ [E] and other gravity aspects (including the**
**combed strike angles with the *Comb* statistics) are shown in S4:25.**



This tail is replicated in the younger relief uplifted SW of the Ticul fault (see ETOPO 1
morphology, Fig. 2a). Extending the trench axis southward (Fig. 2b), another linear depression
(dark and light green) is encountered in a nearly perpendicular direction, trending northward and
forming a "V" like shape. For both these structures, we suggest that the reviving influence of the
impact on pre-existing geological structures may have been at work.

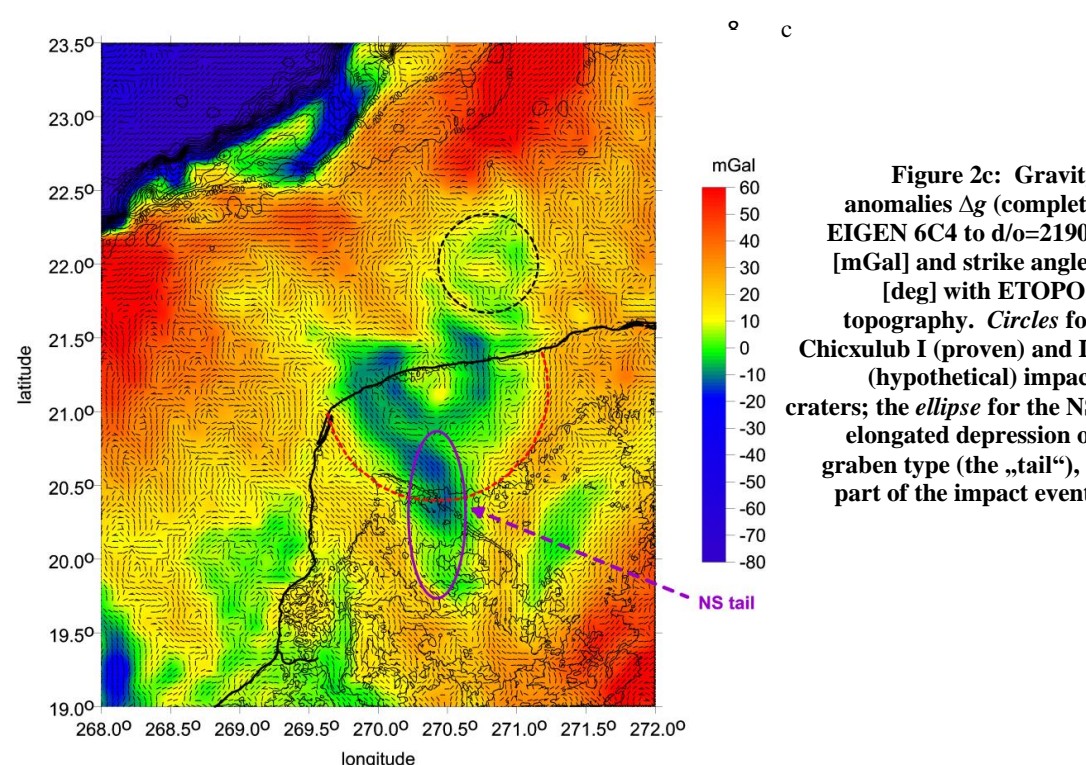

**Figure 2c: Gravity anomalies Δ*g* (complete EIGEN 6C4 to d/o=2190) [mGal] and strike angles [deg] with ETOPO1 topography.** *Circles* for Chicxulub I (proven) and II (hypothetical) impact craters; the *ellipse* for the NS elongated depression of graben type (the „tail"), a part of the impact event.



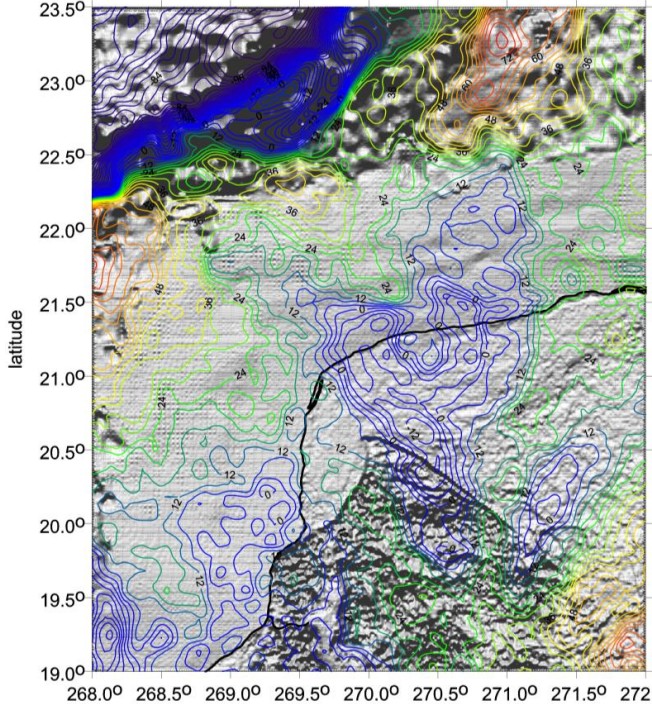

d

**Figure 2d: Gravity anomalies (full EIGEN 6C4) [mGal] as contour lines and the ETOPO 1 topography [m]as shaded relief. For more figures see S4: 5, 11, 12, 19, 20, and 28. Negative values of $\Delta g$ [mGal] are in blue colour.**

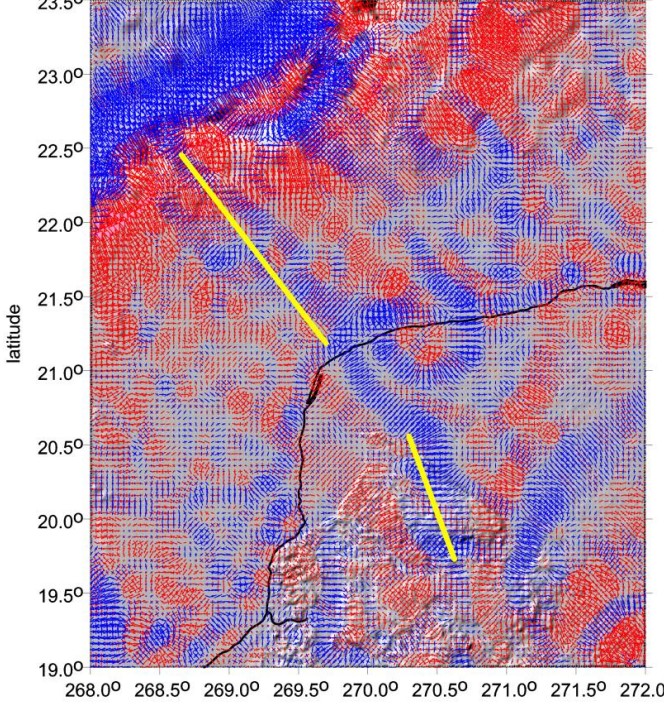

e

**Figure 2e:**

**The virtual deformations $vd$ [-] (compression in blue, dilatation in red) with EIGEN 6C4 in 4 km grid. Black line: the coast. Yellow lines: hypothetical impact graben including the NS „tail" on its southern end.**



## 7. Discussion

*Popigai*

1. Beside the main, proven crater, we clearly see more candidates for the impact craters; they are lined in the NW-SE direction (as we observed in Klokočník et al., 2010 and denoted them there Popigai II, III, and IV). Here we confirm these our previous findings (Figs. 1b-d and S3). The area SE of the main crater has negative values of $\Delta g$ and $T_{zz}$, and aligned strike angles $\theta$.

2. Topography (ETOPO 1) and the gravity aspects do not correlate well. This indicates a partial smoothing of the impact features by erosion and filling of the impact-made depressions, in this case of both craters (including the hypothetical Popigai II crater) and the hypothetical in the NW-SE direction running impact trench.

3. The strike angles are combed into a halo around the main, proven crater Popigai I and are partly overlapping with the aligned but fragmented strike angles for Popigai II (Figs. 1 b,c and S3).

4. A long, EW oriented structure, visible with ETOPO 1 (Fig. 1a) and with the gravity aspects (Figs. 1b,c) is located north of the main crater Popigai I (see the arrows at Fig. 1a). We have mentioned in Sect. 5.1. that unexpectedly the NS faults dominate in the crater without any apparent influence of the impacting body (Masaitis 2008), while other tectonic schemes (Mashchak & Naumov 2005) found the evidence of the expected radial tectonics. Masaitis (1998) in his diagram of the crater shows radial tectonics in the immediate vicinity of the crater, In Vishnevsky & Montanari (1999), however, long faults of NW-SE and SW-SE direction are displayed.

    The NW-SE linear structure connecting the craters is of particular interest to us because in the SE from Popigai I we can observe a long and wide depression to the distance of ~400 km (Figs. 1b,c). This type of image is repeatedly encountered in most geological interpretations of the gravity data, typically, e.g., for the ancient Nile valleys or lake basins covered, e.g., by Saharan aeolian sands, or hidden under the Antarctic glaciers. We therefore assume that a depression filled with younger sediments extends south of the Popigai craters. According to analogies with other terrestrial structures, the thickness of the fill could be 1 km or more.

5. Given the close spatial association of the circular impact structure (the crater) to the linear NW-SE running „basin", we guess the linear feature can be original tectonic belt that was reactivated in extensional mode after the impact and subsequently filled with sediments in a dynamically





evolving Cenozoic landscape. It could be formed or influenced by Neogene movements related to the Tethys belt, but also to the periglacial regime of the Siberian North. Long-term evolving terrains always have a complex tectonic framework, or rather a sequence of tectonic regimes creating a network of faults of different ages and orientations.

Looking at the broader tectonic framework, one can see a number of significant structures based on faults of roughly NS or NW-SE direction. It follows the Ural Mountains, the western margin of the Central Siberian Plateau, the Verkhoyansk Chrebet, and other rivers such as the Daldyn River directly in the crater and around. The NW-SE linear depression resembles an impact graben, i.e. a „trench modified by impact". The basis of an impact graben is a pre-impact geological structure, activated by the impact energy to form a graben. This is not a new concept, as we observe basaltic rock eruptions in the extensional pressure regime in impacts on the Moon and Mars, or in some large terrestrial craters (Sudbury). In contrast, in the compressional tectonic regime, impact horsts are formed, such as those observed on the uplifted crater rims. The two stress-release or compression-extension regimes are complementary, usually perpendicular or oblique to each other. Especially in inhomogeneous terrestrial conditions (except perhaps in stable Archean blocks), meteorites strike areas with already existing regional stress fields. The stresses are then activated in specific directions by the enormous kinetic energy of the impactor.

Mashak and Naumov (2005) stated: "…Thus, the 35-Ma-long post-impact modification history of the Popigai crater is determined by the superimposition of the regional tectonics on the long-term relaxation movements. As a whole, the late modification stage tectonics is found to have only an insignificant effect to the Popigai crater, so that both the original structure and the crater topography have been retained in a good state." The existence of circular structure Popigai II and closely associated trench evokes the possibility of almost concurrent formation of impact crater (or craters) and impact NW-SE graben.

*Chicxulub*

1. Topography (ETOPO 1) and the gravity aspects (namely $\Delta g$, $T_{zz}$, $vd$, and the invariants) do not correlate.
2. The majority of cenotes agree with the innermost ring (or the second ring, when counting the central ring around the central peak as the first) having positive $\Delta g$, $T_{zz}$ and strike angles combed into a halo.



3. The "southern tail" with negative $\Delta g$ and $T_{zz}$ and with the strike angles $\theta$, aligned in the SN direction, seems to be an inseparable part of one impact event (this impact may consist of several explosions). The strike angles, continuing from the main crater from its halo to south, have a stream flowing from the halo to the SN tail, changing slowly its direction from NW-SE to NS; it is looking like one common feature (the crater and the tail together).

4. Besides the main, proved crater, we have predicted earlier (in Klokočník et al., 2010) another smaller crater in NE direction. Accounting for all new gravity aspects, this still remains possible (see circle in Fig. 2c). Moreover, after a careful inspection, one can distinguish several more, small circular features (in Fig. 2b) near the Chicxulub crater (namely SW of it), which might be also impact craters, scattered around the primary. But this is just a speculation.

Christensen et al. (2009) and others argued that the gravity signal near Chicxulub is associated rather with pre-existing Cretaceous basin proposed for this location (Gulick et al., 2008) than with additional crater(s). Our tools ($\Delta g$ and $T_{zz}$) and EGM2008 (predecessor of EIGEN6C4) in 2010 were not sufficient to solve the problem. Moreover, we always wish to rely upon additional geological, geophysical, and other data, when available. Meantime, with the gravity aspects, our tools improved and our experience with the gravity aspects increased. It is specifically the strike angle $\theta$ that proved to be very inspiring for diverse geoapplications in the case that stresses are present. The combed strike angles around Chicxulub create a halo (which is expected and usual phenomenon for the impact craters and basins, similarly as on the Moon or Mars), from which on its south side, a flow of $\theta$ is changing its direction to south. There is no interruption, no jump, no separation as we should observe between two separate geological features, telling us that the crater itself and its southern tail belong to one and the same body.

Previous studies have suggested assymetries in the Chicxulub crater (e.g. Hildebrand et al., 2003; Gulick et al., 2008). This might be used to estimate the direction of impactor in the atmosphere. However, seismic data show significant variations on the composition of the target rocks around the impact site. It is unclear as to whether the angle of impact or target material heterogeneity is responsible for the asymmetry (e.g., Collins et al., 2008).

Similarly as for the Popigai family, here at the Chicxulub crater, we are interested in the linear structures near Chicxulub, namely in a trench-like structure running NW-SE of the main Chicxulub crater (Fig. 2e). It is replicated in younger relief uplifted SW of the Ticul fault (see ETOPO



morphology. Fig. 2a). Extending the trench axis southward in Fig. 2b, another linear depression
(dark and light green) is encountered in a nearly perpendicular direction, trending northward and
forming a "V" like shape. For both of these structures, we suggest that a reviving influence of the
impact on the pre-existing geological structures may have been at work.

5       Similarly to the Popigai crater family in Figs. 1a-d, we can see here in Figs. 2a-e how the

circular impact structure is followed by a tectonic trench. In both craters, its direction roughly
corresponds to the orientation of the surrounding geological structures. Thus we assume that faults,
fault zones or generally weakened structures already existed in these places before the impact.
According to the gravity aspects, where the crater and the adjacent trench have a similar signal,
we believe that the impact activated these structures form what we call an impact graben. However,
both craters were rapidly filled with younger sediments, thus burying both the circular impact
structure and the linear trenches.
Another interesting view is offered by Fig. 2e, which shows the virtual deformation. Let us
focus on the broad, blue lines that emanate from both arms of the crater to the NW. At the
easternmost line, we observe a continuation along the shelf towards the edge of the continental
slope, giving the impression of a valley formed on some impact-weakened zone. The western (blue
marked) zone is much longer, partially overlapping with the structures shown earlier (Figs. 2
b,c,d).
For both crater formations, we consider the existence and a relationship between their circular
(crater) and linear components (graben-like structures). However, there is a different post-impact
geological evolution for the linear trench-like structures, as they naturally become erosional
pathways and as such, they are subject to both down cutting into the bedrock and filling with
younger sediments, in different ways in both locations (Siberia, Yucatan).
**8.  Conclusion**
We confirm and extend our results from Klokočník et al. (2010) which were based on analysis of
$\Delta g$ and $T_{zz}$ derived from the gravity model EGM2008. Now we work the gravity aspects (including
those $\Delta g$ and $T_{zz}$) and with the EIGEN 6C4 model. Thus, we have (in comparison with 2010) better
tools (the set of the gravity aspects) and better model (EIGEN6C4, with global gradiometric GOCE
data). In turn, we are able to support or reject our older results with a higher reliability, with more
weight. The result is that we argue in favour of double/multiple craters – and bring further findings.





The impact affects or creates not only the circular structures but also other accompanying
phenomena. These may be oriented concentrically as the cenote belts, but also as linear trenches
suggesting the existence of the impact grabens. Their orientation and course depend on the regional
tectonic architecture and stress fields prevailing at the time of the impact event and after it.
*Popigai* (Figs. 1a-d) is probably a multiple crater, catena (the smaller craters are located SE of
the main, proven crater). We consider at least Popigai II as proved crater by our new method and
data. SE-NW is the probable direction of the impactor. The broad and long negative gravity
anomaly in SE direction of the main crater Popigai I indicates a close coupling between the circular
impact structure and a linear depression. They are two possibilities: 1. The depression was formed
by reactivation of older geological structures and is the impact graben. 2. The circular structure
adjacent to the Popigai I crater in SW gives impression of another, perhaps shallower and more
erosion-smoothed impact crater, Popigai II. The gravity aspects at least partially suggest the
possibility of a phenomenon that is uncommon on the Earth -– the impact graben may actually
represent a catena.
*Chicxulub* (Figs. 2a-e) is probably a double crater; the smaller crater is located NE of the main,
proven crater. NE-SW is the probable direction of the impactor. The southern negative anomaly
(the tail) belongs to the impact, as clearly demonstrated by the alignments of the strike angles and
changes in their direction. The strike angles are combed into halos around the main crater (typical
situation for all bigger impact craters) but then, on the southern side of Chicxulub I, they turn to
south (creating the tail). This tail can be the most southern end of the impact graben (Fig. 2e)
running NW, W to SW of Chicxulub I in the NW-SE direction.
**Keywords:** impact craters Popigai, Chicxulub – gravity aspects with EIGEN6C4 – gravity strike angles –
multiple craters – impact grabens
**Funding.** This work was supported from the projects RVO: 67985815 and RVO: 67985831 (Czech
Academy of Sciences, Czech Republic).
**Declaration of interests.** The authors declare that they have no competing interests.
**Code/Data availability.** The gravity field parameters of EIGEN6C4 and ETOPO are generic. Our gravity
aspects, computed and plotted by our software, are available on request.
**Author contributions.** All the authors contributed to the data analyses, writing the manuscript, the
discussion and interpretation. Figures were plotted by *surfer* software by Jan Kostelecký.

**Appendix/Supplements.** Supplementary data to this article can be found online at
https://www.asu.cas.cz/~jklokocn/Popigai_Chicxulub_2024_supplements/.



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
