# Peer review of "Popigai and Chicxulub craters"

_EGUsphere, 2024_

## Author Comment (AC1)

**Supplementary material   SM 1**

**THEORY**

*Summary of formulae of the gravity aspects*

The theory is mostly from Pedersen and Rasmussen (1990), Beiki and Pedersen (2010) and from our own papers/books Kalvoda et al. (2013) or e.g. Klokočník et al. (2017, 2020). These last two are the source for this *Supplement*. Examples follow in the second Supplement. References are in the main text and in (Klokočník et al. 2020).

The *disturbing static global gravitational potential* outside the masses of a celestial body (planets, moons) in the spherical harmonic expansion is given by

$$T(r, \varphi, \lambda) = \frac{GM}{r} \sum_{l=2}^{\infty} \sum_{m=0}^{l} \left(\frac{R}{r}\right)^l \left(C'_{l,m} \cos m\lambda + S_{l,m} \sin m\lambda\right) P_{l,m}(\sin \varphi), \tag{A1}$$

where *GM* is a product of the universal gravitational constant *G* and the mass *M* of the planet (also known from satellite analyses as the geocentric gravitational constant in the case of the Earth), *r* is the radial distance of an external point where *T* is computed, *R* is the radius of the planet (which can be approximated by the semi-major axis of a reference ellipsoid), $P_{l,m}$ *(sin φ)* are the Legendre associated functions, *l* and *m* are the degree and order of the harmonic expansion, *(φ, λ)* are the (planeto)centric latitude and longitude, and $C'_{l,m}$ and $S_{l,m}$ are the *harmonic geopotential coefficients* (also known as *Stokes parameters*); fully normalized, $C'_{l,m} = C_{l,m} - C^{el}_{l,m}$, where $C^{el}_{l,m}$ belongs to the reference ellipsoid. The word "*disturbing*" here means the difference between the total gravitational potential of the actual body and the gravitational potential of a reference body, i.e. the reference ellipsoid, usually taken as a rotational ellipsoid with some flattening on the poles due to the rotation of that body.

A set of numbers $C'_{l,m}$ and $S_{l,m}$, presented to a maximum degree $L_{max}$, is called the *gravity/gravitational (field) model of the Earth*.  ("Gravity"

means gravitational effect plus effect of centrifugal force of the studied body).

There is $l \times (l–1)$ terms in such a model, if is complete to the maximum degree and order $l$, $m$ (or $d/o$) and if a few first (lowest degree) terms are not omitted (sometimes these terms are set at zero, due to reasons which will not be discussed here).

The gravity/ gravitational models are usually based on a great amount of diverse satellite and terrestrial data collected from around the world over a long time; then such a result is known as a high-resolution "*combined model*" (e.g., GEM 2008, Pavlis et al. 2012; EIGEN 6C4, Förste et al. 2014), for references see Förste et al. (2014) in contrast to "*satellite-only models*".

Let us recall that $C'_{l,m}$ and $S_{l,m}$ are considered to be constants (excluding a few lowest degree zonal harmonics, which have often been published with a secular trend and semi/annual or other time variable components). We speak about *static gravity/gravitational models.*

There are also *variable gravity/gravitational field solutions*, derived from the global satellite data (mainly from the GRACE mission). They are based on short arc solutions (from observations gathered for one month or a shorter interval), so they are available for a much lower $L_{max}$ than the static models (say to $d/o$ = ~100 instead of ~2000), and only for the Earth.

The *gravity/gravitational aspect* is a functional/function of the gravity gravitational) field potential $T$. It can be its derivative or any other function, often non-linear. We work with the following *gravity aspects* (descriptors): the gravity anomaly (or disturbance) $\Delta g$, the Marussi tensor ($\boldsymbol{\Gamma}$) of the second derivatives of the disturbing potential ($T_{ij}$), two gravity invariants ($I_j$), their specific ratio ($I$), the strike angles ($\theta$) and the virtual deformations ($vd$) – Klokočník et al. (2017, 2020).

All the gravity aspects together provide thorough information about the density anomaly due to the causative body that is more complete than, for example, information that the traditional and usual gravity anomalies themselves could yield. The set of gravity aspects informs about location, shape, orientation, a tendency to a 2D or 3D pattern, and stress tendencies and may partly simulate "dynamic information" although the input data are always the same – those harmonic geopotential coefficients $C'_{l,m}$ and $S_{l,m}$ of a static gravity field model. The whole theory is arranged in such a way that we cannot use any input other than the harmonic coefficients of a gravity model.

The spherical approximation of the *gravity anomaly Δg* (free air, without any geophysical model) is computed as the first radial derivative of *T* by

$$\Delta g = -\frac{\partial T}{\partial r} - 2\frac{T}{r}. \tag{A2}$$

Instead of (A2), one can use the *gravity disturbance*, which is as (A2), but without the second term (often small). The gravity anomalies/disturbances are computed from measurements by ground, airplane or marine gravimeters or derived from measurements performed by means of satellite altimetry.

The gravity gradient tensor $\mathbf{\Gamma}$ (the *Marussi tensor* or simply the *gravity tensor*) is a tensor of the second derivatives of the disturbing potential *T* of the gravity field model. The Marussi tensor was considered the centerpiece of traditional differential geodesy; up to the second order this tensor systematically synthesizes all the dynamical and geometric properties of the Earth's gravity field (see Klokočník et al., 2020 for references).

The tensor $\mathbf{\Gamma}$ is given in the local north-oriented reference frame (*x*, *y*, *z*), where *z* has the geocentric radial direction, *x* points to the north and *y* is directed to the west (Pedersen and Rasmussen, 1990):

$$\mathbf{\Gamma} = \begin{bmatrix} T_{xx} & T_{xy} & T_{xz} \\ T_{yx} & T_{yy} & T_{yz} \\ T_{zx} & T_{zy} & T_{zz} \end{bmatrix} = \begin{bmatrix} \frac{\partial^2 V}{\partial x^2} & \frac{\partial^2 V}{\partial x \partial y} & \frac{\partial^2 V}{\partial x \partial z} \\ \frac{\partial^2 V}{\partial y \partial x} & \frac{\partial^2 V}{\partial y^2} & \frac{\partial^2 V}{\partial y \partial z} \\ \frac{\partial^2 V}{\partial z \partial x} & \frac{\partial^2 V}{\partial z \partial y} & \frac{\partial^2 V}{\partial z^2} \end{bmatrix} \tag{A3}$$

Outside of the body masses $\mathbf{\Gamma}$ satisfies Laplace's differential equation, i.e. the trace of the Marussi tensor (A3) is zero. The tensor $\mathbf{\Gamma}$ is symmetric ($T_{yx}=T_{xy}$, $T_{zx}=T_{xz}$, $T_{zy}=T_{yz}$) and harmonic ($T_{xx}+T_{yy}+T_{zz}=0$); it contains nine components, but just five linearly independent components.

Gravity gradiometry is the measurements of $T_{ij}$. The gravimeters measure the first derivative of *T*, i.e. the accelerations *Δg*, the gradiometers measure the second derivatives of *T*. From the actual formulae for their computation (elsewhere) one can see that the gravity gradients are more sensitive to the close-by mass distribution (density anomalies) than the gravity accelerations.

Terrestrial gradiometers (torse balances) to measure $T_{ij}$ are known from geophysics (see any textbook), but they were not too successful in practical use (too noisy). Now, owing to technical progress, they can successfully be used for measurements on board of airplanes and are used for prospection at local scales.

The *GOCE* mission (Gravity and [steady state] Ocean Circulation Explorer, ESA) was the first (2009) and until now (2022) still the last gradiometric instrument working successfully in space on low orbit (measuring six of $T_{ij}$ components million times during its ~5-year lifetime at a carefully selected orbit fulfilling special criteria on orbital resonances), based on micro-accelerometers (a pair of them in each spatial direction *x,y,z*).

The Marussi tensor has already been used locally (this means in areas of a few per few kilometers) for petroleum, metal, diamond, groundwater, etc., explorations (for more information and for further references see Klokočník et al. 2020a).

The Marussi tensor is a rich source of information about density anomalies providing useful details about the target objects shallowly located beneath the Earth's surface. This extra information can be used by tensor imaging techniques to enhance the source anomalies; it has been tested for local features (economic minerals, oil and gas deposits, fault location, etc.). The tensor components are used at local scales to identify and map the geological contact information, either the edges of the source targets or the structural/stratigraphic contact information. The horizontal components identify the shape and the geological setting of a responsible body. The quantity $T_{zz}$ is best suited for target body detection; $T_{zz}$ helps to define the isopath/density relationships of body mass with relation to its geological setting, see e.g. Saad (2006).

Under arbitrary coordinate transformation, any gravity field and any **Γ** have just three global *gravity invariants* which remain constant. Here they are labelled $I_0$, $I_1$, and $I_2$:

$$I_0 = trace\ (\mathbf{\Gamma})$$

and this one is zero outside the masses of the studied body (known also as the Laplace equation). The remaining two invariants read in general:

$$I_1 = \tfrac{1}{2}\ [trace\ (\mathbf{\Gamma})^2 - trace\ (\mathbf{\Gamma}^2)],$$

$$I_2 = det\ (\mathbf{\Gamma}).$$

This can be transformed (using the components of $\mathbf{\Gamma}$ in Eq. 3) to:

$$I_0 = T_{xx} + T_{yy} + T_{zz}$$

$$
\begin{aligned}
I_1 &= (T_{xx}T_{yy}+T_{yy}T_{zz}+T_{xx}T_{zz}) - (T_{xy}{}^2+T_{yz}{}^2+T_{xz}{}^2) = \\
&\quad \sum_{\{i,j\}\in\{x,y,z\}}\left(T_{ii}T_{jj} - T_{ij}^2\right) \\
&= -\,(T_{xx}{}^2+T_{yy}{}^2+T_{xx}T_{yy}+T_{xy}{}^2+T_{yz}{}^2+T_{xz}{}^2),
\end{aligned}
\tag{A4}
$$

$$
\begin{aligned}
I_2 &= det\,(\mathbf{\Gamma}) = \\
&= T_{xx}\,(T_{yy}T_{zz}-T_{yz}{}^2) + T_{xy}\,(T_{yz}T_{xz}-T_{xy}T_{zz}) + T_{xz}\,(T_{xy}T_{yz}-T_{xz}T_{yy})
\end{aligned}
\tag{A5}
$$

The invariants are mathematically independent of the coordinate system chosen, so invariant ("resistant") with respect to any rotation. The invariant $I_0$ is useful for numerical checks of the actually measured $T_{ii}$. The invariant $I_1$ is the sum of the six products of two tensor coefficient matrix elements, a nonlinear functional model with regard to the geopotential harmonics. The invariant $I_2$ is the determinant "det" of $\mathbf{\Gamma}$.

The invariants can be looked upon as non-linear filters enhancing sources with big volumes (Pedersen and Rasmussen, 1990). They discriminate major density anomalies into separate units. It is useful and helpful that the resultant computed anomaly response retains the same shape and orientation, i.e. it is independent of the observer's choice of axes; this is significant for interpretation when mapping geological structures.

Pedersen and Rasmussen (1990) showed that the ratio $I$ of the invariants $I_1$ and $I_2$, defined as

$$\le I = -\frac{(I_2/2)^2}{(I_1/3)^3} \le 1\,, \tag{A6}$$

always lies between zero and unity for any potential field. If the causative body is strictly 2D (flat), then $I=0$. Thus, the ratio can be an indicator of two-dimensionality, sometimes called the "2D factor". If $I=0$, then we have the necessary but not sufficient condition for two-dimensionality. If the causative body – as seen from the observation point – looks more "3D-like" (for example, a volcano), then $I$ grow and eventually approach 1.

The gradient tensor $\mathbf{\Gamma}$ contains information about subsurface strike (stress) directions. Pedersen and Rasmussen (1990) defined the *strike angle $\theta$* (strike lineaments, strike direction) as follows:

$$\tan 2\theta_s = 2\frac{T_{xy}(T_{xx}+T_{yy})+T_{xz}T_{yz}}{T_{xx}^2-T_{yy}^2+T_{xz}^2-T_{yz}^2} = 2\frac{-T_{xy}T_{zz}+T_{xz}T_{yz}}{T_{xz}^2-T_{yz}^2+T_{zz}(T_{xx}-T_{yy})} \qquad (A7)$$

where $\theta$ is estimated within a multiple of π/2; and only one value represents the main direction of **Γ.** Provided that the ratio $I$ in (A6) is small, the strike angle may indicate a dominant 2D structure. If one were able to rotate with the structure in such a way that the elements of the first row and first column of **Γ** were identically equal to zero, then one would reach a "correct" direction of "stress fields" described by $\theta$ (Beiki and Pedersen, 2010).

Mathematically, $\theta$ is the main direction of **Γ**. Geophysically, it is an important direction for the ground structures; it may indicate areas with a lower porosity or "stress directions".

The strike angles usually show chaotic directions. Sometimes, they are oriented dominantly in one prevailing direction (linearly or creating a halo around the object), they are aligned, combed. The combed values, mostly for small $I$ (*I<0.3*), may signalize possible oil or gas fields, ground water, paleolakes or impact craters (e.g., Klokočník et al. 2020, and further references there).

The situation remains, however, not unambiguous when solely using the gravity data. The reason is that not only can oil and gas fields be detected by the combed $\theta$ but also groundwater reservoirs, water-filled depressions, paleolakes or stress fields after impact at and near the impact craters. The combed $\theta$ probably relates to changes of porosity and stresses, for example due to impact pressure deformations. It is evident that we need additional information to the gravity aspects, geological or geophysical information, namely magnetic anomalies, archaeological data, detailed surface or subglacial topography, etc.

Now let us define the *"virtual deformation"* (*vd*), introduced for the first time by Jan Kostelecký in Kalvoda et al. (2013). It is analogous to the tidal deformation known from geodesy and geophysics; one can imagine the directions of such a deformation due to "erosion" brought about solely by gravity.

If there were a tidal potential represented as in our case by $T$ (A1), then horizontal shifts (deformations) would exist due to this and they could be expressed in the north-south direction (latitude direction) as

$$u_\Phi = l_s \frac{1}{g} \frac{\partial T}{\partial \varphi} \qquad (A8)$$

and in the east-west direction (longitudinal direction) as

$$u_\Lambda = l_S \frac{1}{g\cos\varphi} \frac{\partial T}{\partial \lambda} \qquad \text{(A9)}$$

where $g$ is the gravity acceleration 9.81 m·s$^{-2}$, $l_S$ is the elastic coefficient (called the Shida number) expressing the elastic properties of the Earth as a planet (generally $l_S = 0.08$), $\varphi$ and $\lambda$ are the geocentric latitude and longitude of the point $P$ where we measure $T$; and the potential $T$ is expressed in [m$^2$ s$^{-2}$]. In our case, $T$ is represented by Eqs. (A1), (A8) and (A9). The practical problem is that the actual values of the Shida parameters $l_S$ for the Earth's surface (for the specific locations) are not known; thus, we will know (A8) and (A9) and subsequent quantities only as relative values.

The formalism of continuum mechanics was applied to derive the main directions of the deformations, meaning to transform the horizontal shifts to a small deformation. The tensor of a small deformation $E$ is defined as a gradient of the horizontal shifts (A8) and (A9):

$$E = \begin{pmatrix} \epsilon_{11} & \epsilon_{12} \\ \epsilon_{21} & \epsilon_{22} \end{pmatrix} = \begin{pmatrix} \frac{\partial u_x}{\partial x} & \frac{\partial u_x}{\partial y} \\ \frac{\partial u_y}{\partial x} & \frac{\partial u_y}{\partial y} \end{pmatrix}. \qquad \text{(A10)}$$

The tensor $E$ has two parts: $\mathbf{e}$ is the symmetrical, and $\mathbf{\Omega}$ is the anti-symmetrical part:

$$E = \mathbf{e} + \mathbf{\Omega} = \left(e_{ij}\right) + \left(\Omega_{ij}\right) \qquad \text{(A11)}$$

The symmetrical tensor $\mathbf{e}$ reads:

$$\mathbf{e} = \begin{pmatrix} e_{11} & e_{12} \\ e_{21} & e_{22} \end{pmatrix} =$$
$$\begin{pmatrix} \epsilon_{11} & (\epsilon_{12} + \epsilon_{21})/2 \\ (\epsilon_{12} + \epsilon_{21})/2 & \epsilon_{22} \end{pmatrix}, \qquad \text{(A12)}$$

the parameters of deformation are:

$\Delta = e_{11} + e_{22}$        total dilatation        (A13)

$\gamma_1 = e_{11} - e_{22}$        pure cut

| | |
|---|---|
| $\gamma_2 = 2e_{12}$ | technical cut |
| $\gamma = (\gamma_1{}^2 + \gamma_2{}^2)^{1/2}$ | total cut |
| $\boldsymbol{a} = \frac{1}{2}(\Delta + \gamma)$ | major semi-axis of the ellipse of deformation |
| $\boldsymbol{b} = \frac{1}{2}(\Delta - \gamma)$ | minor semi-axis of the ellipse of deformation |
| $\boldsymbol{\alpha} = \frac{1}{2}\operatorname{atan}(\gamma_2/\gamma_1)$ | direction of the main axis of deformation. |

Note that different specialists make use of different terminology for the same or similar quantities quoted in (A13).

The derivatives of the disturbing geopotential obtained originally as the directional values (A8), (A9) were transformed to small positional deformations, or shifts. But basically, the information content of the Marussi tensor and of *vd* is the same, only exposed in different ways.

To illustrate *vd*, the semi-axes *a*, *b* of the deformation ellipse are computed. As we already know, the local values of $l_S$ are not known, and, in turn, only the main directions of *vd* (and not their amplitudes) can be computed.

It is very interesting and may sounds unusual that *vd* provide dynamical information, even though they are, as well as all the gravity aspects mentioned here, computed from static gravity models (represented by a set of $C'_{l,m}$ and $S_{l,m}$).

As already mentioned, the *vd* is analogous to the tidal deformation and characterizes the "tensions" (directional compression and dilatation) generated by the causative body (Kalvoda et al. 2013). We can understand the *vd* as a principal axis transformation from the horizontal gradients of the deflections of the vertical (A8) and (A9). Since the potential is forward modelled from the topography (at least in the case of the RET 14 model, see below), it is also related to curvature of topography.

*Notes to the combed strike angles*

The *combed strike angles* are strike angles $\theta$ oriented roughly in one and the same direction. Here we define the combed coefficient *Comb* for $\theta$ as a measure or degree of $\theta$ being combed; it is a relative value in the interval $\langle 0,1 \rangle$, where 0 means to be "not combed" (the vectors of $\theta$ are in diverse directions) and *1* means to be "combed" (perfectly aligned, the vectors of $\theta$ are oriented into one prevailing direction). They are different ways how to arrange such a tool.

The following are the input data to the statistics:

$$\theta_i \in \langle -90°, 90° \rangle, i = 1, \dots, n$$

for *n* pixels in the studied area or zone. We compute the main direction of the combed $\theta$ as the mean value of $\theta_i$; let us denote it as $\theta_{Comb}$:

$$\theta_{Comb} = \frac{\sum_{i=1}^{n} \theta_i}{n}$$

by choosing the angles $\theta_i$ either in the interval $\langle -90°, 0° \rangle$ or in the interval $\langle 0°, 90° \rangle$. We use the following important condition:

$$\forall(|\theta_i - \theta_{Comb}| > 90°): \theta_i = 180° - |\theta_i|$$

which means that even two angles $\theta_i$ in opposite directions are counted as one direction. For example: for $\theta_{Comb} = 80°$ and $\theta_i = -80°$, a deviation from the main *Comb* direction is 20°.

Let us define a root mean square value of scatter (variance) of $\theta_i$ for *n* pixels as:

$$rmsv = \sqrt{\frac{\sum_{i=1}^{n}(\theta_i - \theta_{Comb})^2}{n}}$$

Then the required looked-for value of the main *Comb* direction can be defined as

$$Comb = 1 - \frac{rmsv}{90°}$$

As a measure of the degree of $\theta$ "being combed", we make use of the relative values of $\theta_i$:

$$\theta_i^{relat} = 1 - \frac{abs(\theta_i - \theta_{Comb})}{90°} \qquad (A14)$$

The *Comb* value is shows local direction of the tested set of $\theta_i$ in the given region; the departures of the individual $\theta_i$ from *Comb* are plotted in a preselected optimum size of *n* rectangular pixels in a relative scale; these are the values of (A14); if some of $\theta_i$ fit in the main direction of *Comb*, then the pixel has the value 1; if not, then the values (A14) are in the interval $\langle 0,1 \rangle$. This serves as a simple statistical evaluation to compare areas with combed strike angles to those with "non-combed" strike angles.

If *Comb* is smaller than 0.55, we say that $\theta_i$ of the given region are "not combed"; if *Comb* >0.65, we say $\theta_i$ are "combed". There is a "grey zone" between the two, i.e. *Comb*= 0.55–0.65.

*Note on DATA in EIGEN 6C4*

The most important is the gravity field model used. We make use of a high resolution combined *E*uropean *I*mproved *G*ravity model of the *E*arth by *N*ew techniques (*EIGEN 6C4*, Förste et al. 2014), expanded to degree and order (d/o) 2190 in spherical harmonics; this corresponds to the ground resolution 5x5 arcmin or ~9 km on surface.  Precision of EIGEN 6C4, expressed in terms of $\Delta g$, is 10 mGal, but in many civilized land areas and over the oceans and open seas is much better. The authors of EIGEN 6C4 have not access to most of the recent high resolution terrestrial gravity data on the continents, thus they took a synthesized gravity anomaly grid based on EGM2008 (Pavlis et al. 2012). That means that the errors for high d/o terms in EIGEN 6C4 are dominated by the relevant errors in EGM2008. To estimate the precision for the given area of interest, not only a general figure 10 mGal, one needs to inspect gravity anomaly commission error maps of EGM2008 (Pavlis, reference above, the map below). For the northern Yucatan peninsula, we get 4-8 mGal, for Popigai in Siberia a bit worse.

[Figure]

**References**

Beiki, M., Pedersen, L.B., 2010. Eigenvector analysis of gravity gradient tensor to locate geologic bodies. *Geophysics* 75, 137-149; DOI: 10.1190/1.3484098.

Förste, Ch., Bruinsma, S., Abrykosov, O., Lemoine, J-M. et al., 2014. The latest combined global gravity field model including GOCE data up to degree and order 2190 of GFZ Potsdam and GRGS Toulouse (EIGEN 6C4). *5th GOCE user workshop*, Paris, 25-28 Nov.

Kalvoda, J., Klokočník, J., Kostelecký, J., Bezděk, A., 2013. Mass distribution of Earth landforms determined by aspects of the geopotential as computed from the global gravity field model EGM 2008. *Acta Univ. Carolinae, Geographica* XLVIII, 2, Prague.

Klokočník, J., Kostelecký, J., Bezděk, A., 2017. *Gravitational Atlas of Antarctica*, Series Spring Geophysics. Springer Nature, 113 pp.; ISBN: 978-3-319-56639-9.

Klokočník, J., Kostelecký, J., Cílek, V., Bezděk, A., 2020. *Subglacial and underground structures detected from recent gravito-topography data.* Cambridge SP. ISBN (10): 1-5275-4948-8; ISBN (13): 978-1-5275-4948-7.

Pavlis, N.K., Holmes, S.A., Kenyon, S.C., Factor, J.K., 2012. The development and evaluation of the Earth Gravitational Model 2008 (EGM2008). *J. Geophys. Res*. 17, B04406; DOI: 10.1029/2011JB008916, 2012.

Pedersen, B.D., Rasmussen, T.M., 1990. The gradient tensor of potential field anomalies: Some implications on data collection and data processing of maps. *Geophysics,* 55, 1558-1566.

Saad, A.H., 2006. Understanding gravity gradients – a tutorial, the meter reader. *The Leading Edge*, 941-949. Ed. B. Van Nieuwenhuise, August issue.

---

## Author Response (AR1)

**Rebuttal letter to comments of Referee #1 and Referee #2**

**Answers to comments of Referee #1**

**Reply of the authors of**
**Popigai and Chicxulub craters: multiple impacts and their associated grabens**
**RC1**: 'Comment on egusphere-2024-866', Anonymous Referee #1, 28 May 2024.

Thank you for the detailed review. We are ready to account your critical comments and if possible further comments based on other suggestions. We can make deep changes in the manuscript. We are not experts on the impact craters; these results are **only one example of our applications of the gravity aspects to test various features** on the Earth, the Moon, and Mars (see references in our manuscript). It is well possible that we missed some important references from really a great amount of them, so we would be happy if you are more specific in your review and would recommend what is missing and should be newly mentioned.

The manuscript could be significantly changed if we get a chance to publish it.

(1) We guarantee our computations of the gravity aspects, but we cannot be sure about our new interpretations. You are expert, we cannot compete.

(2) You wrote: *…a fatal flaw of not constraining their results with other available data that would allow for additional constraints and in this case refute their ideas of multiple impacts in the region of either of these craters.* Based on the seismic profiles, you conclude (a majority opinion) that the gravity low NE of Chicxulub or the southern tail are pre-impact structures. It looks like we did not explain well our results, namely those following from the strike angles. In a big contrast to the gravity anomalies, showing simply a gravity low for NE of Chicxulub or the tail, the strike angles react with a halo as for any other impact crater, not an arbitrary circular-like gravity low.

We know majority opinion that the gravity low NE of the main Chicxulub crater and the southern anomaly (the tail) are pre-impact structures. It would be naive to base our hypothesis that there is more than one crater only on gravity anomalies. But also the strike angles behave as expected for the impact crater (see above and Figs. 2b–e).

(3) We do not work only with the gravity anomalies but with the gravity aspects, a set of functions of the disturbing gravitational field potential. May we recommend the reader to read the theory of the gravity aspects in Klokočník al (2017) and (2020) or in *Supplement* S1 (which was submitted to *SE EGU Discussion* together with the main text)? The difference between the gravity anomalies and the gravity aspects is like the difference between a bicycle and a Cadillac.

(4) Although we now drive a Cadillac and not a bicycle, we are aware that the gravity data alone cannot solve the inverse task in a unique way. We always need additional data. It is emphasized in our text.

We agree with the following critics and would change our text accordingly:

*…Additionally, the discussions of both craters is a strange combination of specific details that are not really important to their regional story and missing other papers and data that could test their proposed ideas…*

We need, however, a more specific review. We would be happy for recommendations of missing papers, not only for a general, critical remarks like … *in both the case of Popagai and Chicxulub the authors have not used their new data to examine exciting details of impact structure and processes where we know there are impacts to study and instead have gone down the proverbial rabbit hole to chase circular gravity lows arguing for impacts where the geologic data in both locations do not show any evidence.*

Would it be possible to take our hypothesis about Chicxulub and Popigai as a working hypothesis, a minority opinion, and test it with a new look? If you assume that a crater is a double/multiple crater, you probably will conduct your research in another way/style than if you stick to the traditional, majority view. We know that against an impact origin of Chicxulub II speak the seismic profiles. But we found only 1-2 such profiles NE of the main Chicxulub crater roughly in the direction where we predict the second crater (the profiles from e.g. Gullick et al 2008, 2016). We do not know about any seismic profile crossing the southern tail. In a big contrast to this, the network of seismic profiles crossing the main Chicxulub crater is dense. For the case of Popigai, we did not find any seismic profile at all. The magnetic field over the Popigai region is based (probably till now) on data from a 2.5 km grid compilation of the former Soviet Union.

Kind regards: the authors.

**RC1**: *'Comment on egusphere-2024-866'*, Anonymous Referee #1, 28 May 2024.

*I was excited to read a paper discussing both Popagai and Chicxulub given these are two of the largest preserved impact basins on Earth. The authors are experts in gravity methods and do a very nice job laying out the efforts to provide update gravity anomaly maps for each of the two impact structures. However, the inferences the authors go on to make about both structures include a fatal flaw of not constraining their results with other available data that would allow for additional constraints and in this case refute their ideas of multiple impacts in the region of either of these craters. Additionally the discussions of both craters is a strange combination of specific details that are not really important to their regional story and missing other papers and data that could test their proposed ideas. For instance at Popagai the whole discussion about diamonds as a shock indicator is unneeded and instead a discussion of the structural geology, erosion depth, drilling results, etc should have been included to allow for aspects like interrogating the new gravity map for what it represents in terms of Popagai's structure. You should a very interesting low gravity ring that could represent the annualar trough for instance or could represent a highly shocked peak ring (an exciting possibility!)...yet instead of diving into such intriguing results, the authors focus on other features in the region that can be explained tectonically and do not have any evidence of an impact origin. For Chicxulub, the authors similarly focus on a detail that is not resolvable in their gravity map (the Holocene aged karst feature the ring of cenotes, which exist due to caves formed within the inner ring of the crater), but then do not detail the ways in which their new gravity map differs from previous ones that might be illustrative of the crater heterogeneities. For instance its a cool opportunity to consider the relative contributions from impact trajectory versus target heterogeneity. Instead they*

*suggest the pre-existing sedimentary basin to the NE of the crater might be a second impact despite seismic images across this feature which show no structure evidence for any kind of impact (no rim faults, terrace zone, central uplift, etc- see the papers already referenced for plenty of evidence its not an impact). Just because it is a gravity low does not mean it's an impact...it usually just means it's a sedimentary basin. Moreover the authors then try to attribute the pre-existing rift that lies south of the crater and likely continued through the area the impact formed as somehow caused by the impact. This structure has been drilled and imaged on seismic data is a known tectonic graben of Jurassic to Cretaceous age, not made by the Cretaceous-Paleogene impact. In effect the features being highlighted adjacent to Chicxulub are well studied in other papers and do not correlated in age and have data specifically demonstrating these are not of impact origin. Thus in both the case of Popagai and Chicxulub the authors have not used their new data to examine exciting details of impact structure and processes where we know there are impacts to study and instead have gone down the proverbial rabbit hole to chase circular gravity lows arguing for impacts where the geologic data in both locations do not show any evidence. I suggest to reject this paper as it stands and recommend to the authors to rebuild their paper using these gravity anomaly maps to improve our knowledge of Chicxulub and Popagai instead.*

**Answers to comments of Referee #2**

- **RC2**: ['Comment on egusphere-2024-866'](), Anonymous Referee #2, 25 Sep 2024.

*Thank you for your careful review. We improved our manuscript accordingly. Technical note: Here as well as in the revised manuscript, the replies, comments and corrections are in blue colour and in italics.*
* * *
The manuscript entitled "Popigai and Chicxulub craters: 2 multiple impacts and their associated grabens" discusses the possibility of Popigai and Chicxulub impact craters being double or multiple craters and the possible reactivation of weak planes in the vicinity of the impact zone resulting in the formation of graben. The authors have conducted a thorough gravitational analysis of the impact craters in great detail using gravity field model EIGEN 6C4 with GOCE gradiometry data. Overall, the volume of work is impressive and the gravitation data expressed in the figures are easily comprehensible. *Thank you.*

However, there are quite a few issues with the discussion and interpretation that needs to be addressed before publication. *Sure, done, the revised text is sent to be posted.*

Major Comments

1. There is a lack of convincing arguments on why the craters are being interpreted as double or multiple craters.

*In the case of Popigai, we guess, our arguments are strong enough (3-4 craters lined-up in ES-NW line). It is not so robust for Chicxulub. Principally, it cannot be decisive only with the aid of the gravity data. More below.*

The authors have very briefly touched upon how double or multiple craters form, which needs to be explained further.

*We guess it is correct to introduce the recent discussion about this topic shortly. Our gravity aspects can, however, say nearly nothing about the formation, meaning whether one or two impact craters will be created by the forthcoming impactor(s). We observe their final gravity record as evolved in long time.*

*Our text has been extended:*
*The gravity aspects cannot themselves decide whether the impactor was a single body or a binary asteroid before its impact on the Earth. Both is possible. As noted above (Sect. 5.1.), there is possibility of break-up of one body (a single asteroid) in the atmosphere or a "flying cluster" of bodies encountering the atmosphere. To create a double crater, components of a binary asteroid should have a big distance (hundred kilometres) because velocity of asteroids in the Solar system is much higher than velocity of a point rotating on the Earth's surface. Thus, close binaries can hit on one and the same place and create one crater only. The binaries with a big distance of their components are difficult to be observed. We do not refer to the relevant literature (e.g., Durda et al., Polishook et al., Pravec et al), because it is out of our main focus.*

There needs to be a discussion on why the smaller craters are not secondary craters but rather formed due to binary asteroids or breakup of asteroids into smaller fragments in the atmosphere.

*Our opinion:*
*On the Earth, with a higher gravity than on the Moon, the situation differs from the Moon. On the Earth, catena is a rare feature. On the Moon, catenae (belts of secondary craters, ejecta) are not exceptional.*
*The ejecta (smaller secondary craters) on the Earth would be formed in another direction than a series of the impact craters originating along-track of the impacting body (bodies). It is always such a „fan" from the impact crater roughly into the opposite direction of the falling asteroid(s).*

*There is an example of Steinheim-Ries double craters in Germany (double crater in a majority of opinions). Flying roughly from W, two craters were generated (the smaller Steinheim first of all and then the bigger Ries) and finally the ejecta (green glass known as moldavits, vltavins) – it can still be found mainly on the territory of SW part of Czech Republic.*

[Figure]

*Figures:* The strike angles at Steinheim-Ries craters (Germany) together with the gravity anomalies computed with the gravity field model EIGEN6C4 (left) and with ETOPO 1 topography (right). Their trend is from ~W to ~E, with a fragmented halo around Ries.

*Link to Supplement SM2:*https://www.asu.cas.cz/~jklokocn//CHIC-POP24_supplements/

*Our text has been modified and extended.*

There is also a lack of explanation pertaining to how the authors are interpreting the impact direction. I feel it is important to address and correlate all these points for a more holistic interpretation on the gravity data and its usefulness in deciphering impact phenomenon in planetary bodies.

*As for the direction of the impactor: sometime we can deduce this direction from the direction of the gravity strike angles (Klokočník et al., 2020b). As a good guide, we offer Steinheim-Ries (S2:18). Geologists know that the impactor(s) came roughly from west, creating first the smaller Steinheim, than the bigger Ries. We can verify it independently using the strike angles; they are combed in the ~WE direction, they are skirting around both craters, creating a halo around Ries (see the figure above). For Popigai, in an analogy, we can expect the impactor coming in ES-NW, producing the small(er) crater(s) first, and the biggest, already proven one, as the last, final (Figs. 1b-d, S3:6-21).*

*A general geological note:*

*Our interpretation is based mainly on the experience with the formation of impact craters on the Moon. In "Atlas of the Gravity and Magnetic Fields on the Moon" (Klokočník et al., 2022), we have studied the gravity characteristics of dozens of impact craters distributed over the lunar surface. Needless to say, lunar craters are not only numerous, but, except for the oldest mascon-type structures, little altered by later processes, whereas on Earth, erosion often results in root-like structures several kilometres deep or, as in the case of Chicxulub, in phenomena buried beneath younger sediments. This means that we have worked mainly with analogies that are additionally obscured by erosional processes on Earth, and the original gravity signal may be overprinted by other processes such as tectonic activity or selective erosion. Therefore, the aim of this paper is not to provide unequivocal evidence for the existence of additional craters, but probability that they exist and that further field research should tell more.*

*In the case of the Popigai impact, the gravitational anomalies are arranged roughly in a single line, which in our opinion best corresponds to lunar catenae. These structures will be difficult to prove on Earth because the smaller craters in particular were formed with much lower impact energy. Thus, we can expect that the impact structures were of varying depths and the shallower ones were more affected by erosion. They will therefore appear in the gravimetric record with different intensities, or they may have disappeared completely. The uniqueness of Popigai Crater, in our opinion, is that the entire linear structure most closely resembles a catena as we know it on the Moon.*

*The situation at Chicxulub Crater is far from clear. Our data suggest the existence of another crater with some non-negligible probability, but it is fair to say that any interpretation is speculative at this stage of the research. However, if field, i.e. borehole, exploration proves its existence and at least partially clarifies the conditions of its formation, we will have more basis for speculating on the nature of the impact. We have already pointed out in the article that the gravity aspects indicate possible presence of the second structure.*

2.  The authors' use of trend and azimuth information is confusing. Some of the trends in the text are hyphenated while others are not. I suggest making all the trend information hyphenated (e.g., N-S, E-W) and azimuth (direction) information non hyphenated (e.g., SE of crater). In that note, I am confused about the SW-SE fault orientation mentioned in the manuscript (pages 8, 14, 19).

*Hopefully improved. Not everywhere clear.*

Minor Comments

*Shortly speaking: we have no objection against these comments, thus we follow your suggestions to correct our text. Thank you.*

Abstract

Increase the line spacing

Line 19: Rewrite "here the impact craters Chicxulub and Popigai." as "The improved techniques were applied to study the impact craters Chicxulub and Popigai in this present research."

Line 21: Both craters are interpreted to be double or multiple craters.

Line 22: Rewrite 'The both crater formations' as 'Formation of both the craters"

Motivation

Line 28: Instead of writing 'In this journal' it is better to refer the paper

Lines 28-31: Complex sentence; break the sentence to simpler sentences for better readability

Lines 32-33: Brief description of double and multiple craters needs to be added with references

Line 15: Remove 'indeed'

Line 16: shook → shock

Line 23: If magnetic intensities have not been studied in this paper, please refer publications where it has been studied

Notes on Theoretical Preliminaries

29: remove 'gravity'

Lines 8-12: Complex sentence; break the sentence into simpler sentences for better readability

Line 24: Put (Comb) within parenthesis

Line 25: "not combed"

Line 29: Rewrite 'can shape a halo' as 'can take the shape of a halo'

Data, computation, and figures

Line 4-5: Refer the theory

Line 10: Rewrite 'have not access to' as 'did not have access to'

Line 14: Rewrite 'not only a general figure 10 mGal' as 'and not only for a general figure of 10 mGal'

Line 16: Mention how worse

Line 19: Mention couple of other measurements within brackets

Line 20:  heights → height

Line 21: Bedmap 2 → Bedmap2

Line 26: Remove the exclamation mark

Line 4: 'corresponding to the ground resolution of 9 km'

Line 19: Any significance of plotting θ in black and white?

Artefacts

Lines 12-13: Rewrite 'correct interpreting the' as 'the correct interpretation of'

Line 17: with → has

Line 19: unbelievable → unrealistic

Line 22: Rewrite 'how well by the data is covered the area of our interest' as 'how well the data has covered the area of our interest'

Line 30: he → the

Line 1: Moons' → Moon's

Line 9: "hidden"

Line 16: "lurk"

Line 17: attack and distort → hamper

Popigai

Line 27: Remove 'it was'

Line 6: Please clarify whether 'it is' is referring the crater or the shield

Lines 6-7: Please add references

Line 10: Rewrite 'their fig. 3a' as '(cf. Fig. 3a in Pilkington et al., 2002)'

Line15: Section 6.1 does not have any "notes" on binary asteroids, only a brief mention

Line 17: quite remotely area → remote area

Line 19-22: Complex sentence, break the sentence into simpler sentences for better readability

Line 24-25: Why mention beforehand what the authors will argue for?

Line 24: Rewrite 'fragmented now due probably to' as 'which is presently fragmented, possibly due to'

Line 25: Rewrite 'not too intensive' as 'not with too much intensity'

Line 27: mark → signature

Line 30: lined → aligned

Line 4: Rewrite 'we had not' as 'did not have'

Line 5: Begin a new sentence from "With them now……"

Chicxulub

Lines 14-15: Please clarify 'external forcing event'

Line 27-28: Add references

Line 2: "The impact……."

Line 8: Rewrite "The literature about the Chicxulub crater is really rich: from Alvarez……" as "The literature about the Chicxulub crater is really rich. To mention a few: Alvarez……"

Line 9: Remove …

Line 11: Remove "This is not a review paper to mention all."

Line 12: Rewrite "in its study" as "in the study of Chicxulub crater"

Line 14: Rewrite "they did not know" as "they were not aware"

Line 3: Remove the

Line 21: Clarify 'strong on land'

Fig. 2a: Point to the semi-circular shadows

Line 5: Remove 'reviving'

Discussion

Line 30: "basin"

Line 9: "trench modified by impact"

Line 12: Refer the following papers-

Wichman, R. W. (1993). Post-impact modification of craters and multi-ring basins on the Earth and Moon by volcanism and crustal failure. Brown University.

Dasgupta, D., Kundu, A., De, K., & Dasgupta, N. (2019). Polygonal impact craters in the Thaumasia Minor, Mars: role of pre-existing faults in their formation. Journal of the Indian Society of Remote Sensing, 47, 257-265.

Zhang, F., Pizzi, A., Ruj, T., Komatsu, G., Yin, A., Dang, Y., ... & Zou, Y. (2023). Evidence for structural control of

mare volcanism in lunar compressional tectonic settings. Nature communications, 14(1), 2892.

Line 30: central peak as the first ring

Line 16: Meantime → In the meantime

**Citation**: https://doi.org/10.5194/egusphere-2024-866-RC2

*The End of rebuttal letter.*
*Jaroslav Klokočník with co-authors*

---

## Referee Report (RR1)

I have found the revised version of the manuscript entitled "Popigai Chicxulub craters : 2 multiple impacts and their associated grabens" satisfactory and in accordance with reviwers' comments. I recommend accepting the article for publication.

---

## Author Response (AR2)

Author's Response

Dear EGU publications team,

We have modified the manuscript according to the comments of Executive editor, namely we rearranged the layout of the figures and we changed their numbering:

*Executive editor decision: Publish subject to technical corrections*
*by Andrea Di Muro*
*Comments to the author:*
*Dear Authors, I thank you for taking into account the suggestions and remarks received during the review process; I agree with the advice of the topic editor; I would just suggest to change the numbering of the figures (Fig. 2 instead of 1b-c-d; Fig. 3, instead of 2a; Fig. 4, instead of 2b etc), in order to facilitate the layout of the final version of the paper and the reading; thank you again for choosing SE to disseminate your research results, best regards, Andrea Di Muro, SE Executive editor*

Kind regards,
Jaroslav Klokočník with co-authors